# Predictors of Impaired Exercise Performance in Patients Qualified for Cardiac Rehabilitation: The Impact of Sex and Comorbidities

**DOI:** 10.3390/jcm14217512

**Published:** 2025-10-23

**Authors:** Małgorzata Kurpaska, Paweł Krzesiński, Małgorzata Banak, Katarzyna Piotrowicz

**Affiliations:** Department of Cardiology and Internal Diseases, Military Institute of Medicine, National Research Institute (WIM-PIB) in Warsaw, Poland, Szaserow Street 128, 04-141 Warsaw, Poland; pkrzesinski@wim.min.pl (P.K.); mbanak@wim.mil.pl (M.B.); kpiotrowicz@wim.mil.pl (K.P.)

**Keywords:** cardiopulmonary exercise test, oxygen uptake, ventilation-to-carbon dioxide output slop, comorbidities, females, Charlson Comorbidity Index

## Abstract

**Background/Objectives:** Exercise capacity and patient prognosis are heavily influenced by comorbidities. However, the specific impact of individual comorbid conditions on objective measures of exercise performance remains insufficiently characterized. The study aimed to identify predictors of reduced physical capacity in patients qualified for cardiac rehabilitation. **Methods:** A single-center retrospective analysis was conducted on 518 patients qualified for cardiac rehabilitation. After excluding 51 post-cardiac surgery patients, cardiopulmonary exercise testing data from 425 patients (316 men, median age 63 years) were analyzed. Comorbidities data, peak oxygen uptake (peak VO_2_), and the ventilation-to-carbon dioxide output slope (VE/VCO_2_ slope) were evaluated. **Results:** A significantly reduced exercise capacity (peak VO_2_ < 70% of the predicted value) was observed in 29.4% of patients, while an increased VE/VCO_2_ slope (≥36) was noted in 20.8% of patients. Univariate logistic regression identified sex, heart failure, valvular disease, peripheral artery disease, diabetes mellitus (T2DM), chronic kidney disease (CKD), Charlson Comorbidity Index (CCI), left ventricular ejection fraction <50%, diastolic dysfunction, and anemia as predictors of both reduced peak VO_2_ and a steeper VE/VCO_2_ slope. Multivariate regression analysis further identified T2DM and CKD as independent predictors of reduced peak VO_2_, while sex, CKD, and CCI were independent predictors of a steeper VE/VCO_2_ slope. **Conclusions:** Among patients qualified for cardiac rehabilitation, patient’s sex, T2DM, CKD, and the CCI emerged as key predictors of reduced exercise capacity. Reduced peak VO_2_ was more commonly observed in men, while women more frequently exhibited a steeper VE/VCO_2_ slope, indicating potential sex-related physiological mechanisms influencing exercise performance.

## 1. Introduction

Reduced exercise capacity is a common manifestation of cardiovascular diseases and other conditions, such as obesity or diabetes, which can potentially influence cardiovascular performance. The patient’s prognosis and exercise capacity are determined by the presence and nature of concomitant conditions [1], with the prognosis considerably worsened by a sedentary lifestyle and little exercise [2]. The effect of multimorbidity on survival rate is commonly assessed with the Charlson Comorbidity Index (CCI) [3], which has proved its prognostic usefulness in patients with such conditions as coronary artery disease (CAD) [4] and pulmonary embolism [5], and those with a history of myocardial infarction [6].

Exercise capacity is determined both by cardiovascular function and the synergy of the respiratory system, skeletal muscles, oxygen-transporting mechanisms, and metabolic processes [7]. A cardiopulmonary exercise test (CPET) helps objectively assess the patient’s exercise capacity by measuring peak oxygen consumption (peak VO_2_), identifying the causes of dyspnea and limited exercise tolerance, and providing parameters of proven prognostic value, such as ventilation-to-carbon dioxide output slope (VE/VCO_2_ slope). Peak VO_2_ is an established indicator of physical exercise capacity and a predictor of mortality [8], whereas VE/VCO_2_ slope, which is a marker of ineffective ventilation, is considered to be primarily a prognostic parameter [9,10,11,12,13]. In patients with CAD, a peak VO_2_ below 12 mL/min/kg for females and 15 mL/min/kg for males was shown to be associated with the highest annual all-cause mortality [8]. The VE/VCO_2_ slope, on the other hand, was demonstrated to be associated with the prognosis in patients with heart failure (HF) or pulmonary hypertension. Studies showed various VE/VCO_2_ slope thresholds (>34, >36, ≥39.3) for increased mortality or risk of hospitalization (≥32.9) in patients with HF [9,10,14]. Peak VO_2_ is a predictor of both the absolute risk of death and the percentage changes in mortality dependent on the age, weight, height, and sex [8]. Conversely, an absolute VE/VCO_2_ slope is interpreted irrespective of other variables [9,10,11,12,13].

There have been no comprehensive data on the association between peak VO_2_ and VE/VCO_2_ slope on one hand and the burden of comorbidities on the other in patients undergoing eligibility evaluation for cardiac rehabilitation. Existing protocols often do not sufficiently consider the impact of comorbidities and sex-related physiological differences on exercise capacity. The challenge remains to design for patients with comorbidities, e.g., diabetes mellitus [14], chronic kidney disease (CKD) [15], or atrial fibrillation [16]. Identifying predictors of reduced exercise capacity (expressed in the form of abnormal prognostic indicators) may be of significance in optimizing cardiac rehabilitation as a form of a holistic, comprehensive approach. Therefore, this has become the purpose of our study.

## 2. Materials and Methods

### 2.1. Study Population

A single-center retrospective analysis was conducted on the patients found eligible for cardiac rehabilitation between February 2022 and March 2024. We analyzed a data registry of 518 patients and excluded 42 patients who did not undergo CPET, irrespective of the reason, and 51 patients after cardiothoracic surgery due to an acute, temporary condition after surgery, e.g., pleural effusion or restricted breathing due to pain in the sternum. The study population selected in this manner was stratified by exercise capacity and VE/VCO_2_. The subgroup with good exercise capacity (n = 298) had peak VO_2_ ≥ 70% of the predicted value, and another, with significantly reduced exercise capacity (n = 124), had peak VO_2_ < 70% of the predicted value. The VE/VCO_2_ slope of ≥ 36 identified patients with ineffective ventilation (n = 87), in contrast to those with a normal index of ventilatory response to exercise (VE/VCO_2_ slope < 36, n = 331).

### 2.2. Clinical Parameters

Our analysis included data from medical records, such as history of comorbidities, symptoms (including reduced exercise tolerance, dyspnea severity classified with the New York Heart Association (NYHA) classification, and chest pain classified with the Canadian Cardiovascular Society angina scoring system), and medications. The patients’ resting blood pressure, resting heart rate, height, and body weight were measured, and the body mass index (BMI) was calculated. Peripheral venous blood samples were tested for routine parameters (hemoglobin, creatinine, urea, serum lipid, and blood glucose levels). Anemia was defined as hemoglobin levels below 12 g/dL [17]. Estimated glomerular filtration rate (eGFR) was calculated with the Cockcroft–Gault formula. The CKD was defined as having at least a 3-month history of low eGFR (<60 mL/min/1.73 m^2^) or an earlier diagnosis.

The CCI was calculated with the use of a CCI calculator [18].

The data on the left ventricular (LV) ejection fraction, LV diastolic dysfunction, and valvular heart disease were obtained from the results of previous inpatient echocardiography (which took place prior to cardiac rehabilitation eligibility assessment, during percutaneous coronary intervention eligibility assessment, or during decompensated HF). Significant valvular disease was considered at least moderate aortic, mitral, or tricuspid insufficiency or stenosis.

### 2.3. Cardiopulmonary Exercise Test

The CPET was conducted with the use of an Ergoselect cycle ergometer (Geratherm Respiratory GmbH; Bitz, Germany) according to an individualized ramp protocol, in which the target load (calculated with the Wasserman formula [19]) was planned to be reached at minute 10 of exercise. The patient maintained a pedaling cadence of 60–65 revolutions per minute. The test was stopped at the onset of symptoms (fatigue, angina, dyspnea) or at the patient’s request. Continuous analysis of respiratory gases (oxygen and carbon dioxide), preceded by gas calibration prior to each test, was conducted with a CORTEX system (Biophysik GmbH; Leipzig, Germany). Patient monitoring during the CPET included continuous 12-lead electrocardiography and automated blood pressure and oxygen saturation measurements during the 1.5-minute rest period just before the test, during exercise, and during the subsequent 6-minute active rest period. Peak VO_2_ (mL/kg/min) was defined as the mean of the highest 30-second VO_2_ values obtained during exercise. The predictive value of peak VO_2_ was calculated with the Hansen formula [20]. For the purpose of our study, a peak VO_2_ of ≥70% of the predicted value was defined as good exercise capacity (combining the categories of normal exercise capacity of ≥85% of predicted and mildly reduced exercise capacity of ≥70% of predicted), whereas a peak VO_2_ of <70% of the predicted value was considered to indicate significant (moderate to severe) reduction in exercise tolerance [21]. The VE/VCO_2_ slope was calculated with linear regression between VE and VCO_2_ in the period between the first minute of exercise and the end of the isocapnic buffering phase (the second anaerobic threshold) [19]. A VE/VCO_2_ slope of ≥36 was considered to be significantly elevated. All patients had provided their written consent to undergo a CPET.

### 2.4. Six-Minute Walk Test

A 6-minute walk test (6MWT) was conducted in accordance with the 2002 guidelines of the American Thoracic Society [22]. The patients were instructed to walk as quickly as they could along a 30 m corridor marked every 5 m. We analyzed the results of the second 6MWT, performed 2–3 days after CPET. The total distance covered during 6 min (6MWTd) was rounded to 2 m. The obtained absolute values were expressed as a proportion of the predicted 6MWTd values, calculated individually for each patient based on the reference equations introduced by Enright [23].

### 2.5. Statistical Analysis

Statistical analysis was performed using Statistica software, version 12.0 (TIBCO Software Inc., Palo Alto, CA, USA). After normality inspection (Shapiro–Wilk test), the quantitative variables were presented as medians and interquartile ranges (IQRs), while categorical variables were presented as numbers and percentages. Differences between subgroups stratified by exercise capacity (peak VO_2_ <70% vs. ≥70% of predicted) and ventilatory efficiency (VE/VCO_2_ slope ≥36 vs. <36) were assessed using the Mann–Whitney U test for continuous variables and the chi^2^ test for categorical variables.

Univariate logistic regression models were used to identify variables associated with significantly reduced peak VO_2_ and elevated VE/VCO_2_ slope. The most representative variables with a *p*-value < 0.100 in univariate analyses were considered for multivariate models. Collinearity diagnostics (VIF, variance inflation factors) were performed for all candidate variables before entering them into multivariate models. Multivariate logistic regression was performed to determine independent predictors of both reduced peak VO_2_ and steeper VE/VCO_2_ slope. The results were expressed as odds ratios (ORs) with 95% confidence intervals (CIs). A two-tailed *p*-value < 0.05 was considered statistically significant.

## 3. Results

### 3.1. Baseline Characteristics

Exercise capacity was assessed with the CPET in 425 patients (including 316 men), whose median age was 63 years (56–70 years). The most common comorbidities were CAD (98.1%), including status post percutaneous coronary intervention (PCI, 97.4%), non-ST-elevation myocardial infarction (47.8%), and ST-elevation myocardial infarction (24.7%)—Figure 1.

The median peak VO_2_ was 17 (14–22) mL/min/kg, which constituted 78% (67–89%) of the predicted value. The median VE/VCO_2_ slope was 31.2 (27.6–35.3). A total of 281 patients (66.6%) had peak VO_2_ ≥ 85% of predicted value. The median (Q1–Q3) predictive value of 6MWT distance was higher than the median predictive value of VO_2_ (99.8% vs. 78%, respectively). On the other hand, fewer patients had a 6MWT ≥ 100% (208) than had a VO_2_ > 85% (281). A significantly reduced exercise capacity was observed in 29.4% of patients, while an increased VE/VCO_2_ slope was noted in 20.8%—Table 1.

The median 6MWT distance was 555 m (488–611 m), 99.8% of predicted, with 208 patients (50.4%) achieving a 6MWT distance above the predicted values—Table 1.

### 3.2. Comparison Between Patients with and Without an Increased VE/VCO_2_ Slope and Between Patients with and Without Significantly Reduced Peak VO_2_

In comparison with patients with a VE/VCO_2_ slope of <36, the subgroup with a higher VE/VCO_2_ slope comprised a greater proportion of males (*p* = 0.007), presented lower systolic pressure (*p* = 0.015), higher resting heart rate (*p* < 0.0001), lower LV ejection fraction (*p* < 0.0001), lower hemoglobin level (*p* = 0.002), lower eGFR (*p* = 0.0002), and higher CCI (*p* < 0.0001). A higher proportion of patients with a steeper VE/VCO_2_ slope reported more severe dyspnea (higher NYHA class, *p* < 0.0001), HF (*p* < 0.0001), LV ejection fraction < 50% (*p* < 0.0001), LV diastolic dysfunction (*p* < 0.0001), CKD (*p* < 0.0001), valvular disease (*p* < 0.0001), T2DM (*p* < 0.0001), anemia (*p* = 0.003), and peripheral artery disease (PAD, *p* = 0.003)—Table 2.

In comparison with patients with a peak VO_2_ ≥ 70%, the subgroup with VO_2_ of < 70% comprised a greater proportion of males (*p* = 0.014), presented lower systolic pressure (*p* = 0.008), lower LV ejection fraction (*p* < 0.0001), lower hemoglobin level (*p* = 0.042), lower eGFR (*p* = 0.0003), and higher CCI (*p* = 0.0004). A higher proportion of patients with a steeper VE/VCO_2_ slope reported more severe dyspnea (higher NYHA class, *p* = 0.019), HF (*p* < 0.0001), LVEF < 50% (*p* < 0.0001), LV diastolic dysfunction (*p* = 0.002), CKD (*p* < 0.0001), valvular disease (*p* < 0.0001), T2DM (*p* = 0.0008), anemia (*p* = 0.019), and peripheral artery disease (PAD, *p* = 0.003)—Table 2.

Significantly reduced exercise capacity (peak VO_2_ < 70%) was detected in 20.2% of women in comparison with 32.6% of men (*p* = 0.0014), and a VE/VCO_2_ slope ≥ 36 was detected in 29.9% of women in comparison with 17.7% of males (*p* = 0.007). It indicates that women present more frequently with impaired ventilation, while less frequently with poor exercise capacity.

### 3.3. Predictors of Increased VE/VCO_2_ Slope

The risk of a steeper VE/VCO_2_ slope was nearly three times higher in women than in men (1/0.36 = 2.8). Another independent predictor was CKD (OR = 3.6) and the CCI score (OR = 1.5 times per point)—Table 3, Figure 2. A univariate logistic regression analysis also identified other factors as predictors of a steeper VE/VCO_2_ slope: HF, valvular disease, PAD, T2DM, LV ejection fraction <50%, LV diastolic dysfunction, and anemia.

### 3.4. Predictors of Significantly Reduced Peak VO_2_

A univariate logistic regression analysis identified the following variables as predictors of reduced peak VO_2_: HF, valvular disease, PAD, T2DM, history of CKD, CCI score, LV ejection fraction <50%, LV diastolic dysfunction, and anemia—Table 3. A multivariate logistic regression analysis showed T2DM (OR = 2.3) and CKD (OR = 3.2) to be independent predictors of low peak VO_2_—Figure 3.

## 4. Discussion

In this single-center cohort of patients qualified for cardiac rehabilitation, we found that nearly one-third of individuals exhibited significantly reduced exercise capacity, while over one-fifth presented with abnormal ventilatory efficiency. Importantly, impaired exercise performance, as reflected by reduced peak VO_2_ and elevated VE/VCO_2_ slope, was associated with a cluster of comorbidities, including heart failure, diabetes mellitus, chronic kidney disease, and anemia, as well as a high burden of comorbidity quantified by the CCI. Notably, while men were more likely to exhibit reduced peak VO_2_, women more frequently presented with ventilatory inefficiency, suggesting sex-related physiological differences in cardiopulmonary adaptation.

Our study group of patients eligible for cardiac rehabilitation comprised mostly patients with CAD (98.5%), nearly half of whom had been diagnosed with HF (46.1%). Over half of our study population (52.9%) reported exercise intolerance. As the reported symptoms were being verified via CPET, nearly one-third (29.4%) of patients exhibited substantially (moderately or severely) reduced exercise capacity, with an increased VE/VCO_2_ slope (ventilatory class ≥ 2) in one-fifth (20.8%) of patients. Several studies have shown correlations of distance 6MWT with peak VO_2_ in CPET [24]. On the other hand, the discrepancy in the number of patients who achieve >100% and the median predictive value of 6MWT and VO_2_ emphasizes that these tests cannot be treated interchangeably. Given the high burden of cardiac disease, frequent presence of exercise intolerance, and the objective verification by CPET, this cohort represented a clinically relevant and appropriate population for evaluating the determinants of reduced functional capacity.

### 4.1. Exercise Capacity Parameters and Sex

Patients’ sex was a predictor of low peak VO_2_ and high VE/VCO_2_ slope values in univariate analysis and of high VE/VCO_2_ slope values also in multivariate analysis. Importantly, the distribution of sex differed between these two subgroups, with reduced peak VO_2_ more commonly observed in men and an elevated VE/VCO_2_ slope more frequently seen in women. This is consistent with previous reports, which indicated a relationship between the VE/VCO_2_ slope and both sex and age in a population of healthy individuals [25,26,27] and in individuals with HF [26,28]. It was demonstrated that the VE/VCO_2_ slope was steeper by approximately 2 degrees in healthy women than in healthy men [26,28].

The VE/VCO_2_ slope is an expression of both cardiovascular and respiratory function [28,29] and it illustrates ventilatory efficiency [28]. The slope becomes steeper when minute ventilation increases disproportionately to the amount of exhaled CO_2_, for instance, in situations where there is a diffusion barrier for CO_2_ in the form of pulmonary congestion or in cases of hyperventilation. Women have been reported to exhibit better exercise capacity than men despite having a steeper VE/VCO_2_ slope, a finding that did not necessarily correlate with poorer prognosis [30,31,32]. We hypothesize that in women, the steeper VE/VCO_2_ slope is associated with hyperventilation, for example, triggered by emotions or anxiety. In men, the VE/VCO_2_ slope reflects the cause of reduced peak VO_2_. Further research in this area is necessary.

It is worth noting that low peak VO_2_ values may coexist with an elevated VE/VCO_2_ slope, which was demonstrated in earlier reports [33,34]. In our study, the features related to low peak VO_2_ and a steep VE/VCO_2_ slope were comparable: more pronounced dyspnea (NYHA class II and III), higher prevalence of heart failure, LV systolic and diastolic dysfunction, chronic kidney disease, type 2 diabetes mellitus, anemia, peripheral artery disease, and valvular heart disease, as well as elevated comorbidity burden reflected by higher CCI scores. Also, consistent with other expert reports [35], we demonstrated that patients with a steeper VE/VCO_2_ slope have a higher resting heart rate.

### 4.2. Comorbidities Burden as a Determinant of Exercise Capacity and Ventilation

The group of patients evaluated in our study was characterized by high rates of multimorbidity, with the median CCI score of four points, which yields an estimated 10-year survival of 53% [18]. The higher the CCI score, the greater the risk of developing LV systolic and diastolic dysfunction and HF and, consequently, low exercise capacity and a steeper VE/VCO_2_ slope.

Martens et al. reported a relationship between an elevated VE/VCO_2_ slope with low peak VO2 and comorbidity in people with HF [1]. Those authors also observed different prognostic effects of peak VO_2_ and VE/VCO_2_ slope depending on the type of comorbidities. The highest mortality was reported in a subgroup with low peak VO_2_ (13.9 ± 4.1 mL/min/kg) and the steepest VE/VCO_2_ slope (35.8 ± 9.3), whereas no correlation with mortality was observed in patients with both a non-elevated VE/VCO_2_ slope (34.1 ± 8.8) and the lowest peak VO_2_ (13.3 ± 4.01 mL/min/kg) [1].

Common comorbidities affecting cardiovascular patients (e.g., obesity, T2DM, and CKD) damage internal organs, including the heart and blood vessels. In our cohort, patients with a low peak VO_2_ and an elevated VE/VCO_2_ slope had higher rates of low eGFR, CKD, PAD, low hemoglobin levels, and anemia. A steeper VE/VCO_2_ slope was shown to be more likely in patients with concomitant anemia, which—although not directly associated with LV systolic or diastolic dysfunction—has been documented to be relevant in patients with HF [35].

Left ventricular diastolic dysfunction and at least moderate valvular disease (mitral or tricuspid insufficiency) were predictors of both a peak VO_2_ of <70% of predicted and a VE/VCO_2_ slope of ≥36. The relationship between either LV diastolic dysfunction [36,37] or mitral insufficiency [38] and a steeper VE/VCO_2_ slope has been attributed to lower cardiac output and increased pulmonary artery wedge pressure on exertion, which adversely affected exercise capacity and peak VO_2_. Exercise-exacerbated LV diastolic dysfunction is of great clinical significance in HF with preserved ejection fraction manifesting with ventilatory insufficiency, not demonstrated in assessments at rest [35].

We also observed T2DM to be an independent risk factor for significantly lowered peak VO_2_ (OR 2.29; 95% CI 1.15–4.56; *p* = 0.018). This may be caused by, i.e., myocardial systolic dysfunction or impaired oxygen extraction by skeletal muscles [33,34]. Moreover, as reported previously, patients with T2DM and reduced exercise capacity (peak VO_2_ < 80%) present a lower increase in global longitudinal strain (GLS) assessed echocardiographically during exercise (1.9 ± 2.5% vs. 5.9 ± 4.1%, *p* = 0.004) [39].

Although different position statements allow for planning cardiac rehabilitation training based on different parameters [40], our results indicate the usefulness of CPET assessment, especially in patients with T2DM, CKD, and multimorbidity. Designing training using methods other than CPET oxygen thresholds may lead diabetic patients to exercise at too high an intensity, promoting adverse effects [14]. A different chronotropic response in patients with atrial fibrillation also indicates that CPET is more useful than heart rate assessment [16]. Reduced peak VO_2_ and a steeper VE/VCO_2_ slope associated with metabolic abnormalities and overhydration in CKD also highlight challenges in developing cardiac rehabilitation training. The best results in improving performance in CKD were obtained by conducting CR based on the second oxygen threshold also [15].

### 4.3. Limitations

Our study was retrospective in nature, and the analysis was based on data from a single center. Moreover, echocardiography assessments were performed at various labs by cardiologists of varied experience levels, who did not follow a homogeneous examination protocol. Some of the evaluated records were incomplete; however, this was so rare that the risk of it affecting the results of this study is relatively low. Due to the lack of data, it was impossible to assign three patients to the comparative analysis according to peak VO_2_ and seven patients to the analysis according to VE/VCO_2_ slope.

We observed no differences between the peak VO_2_ and VE/VCO_2_ slope subgroups in terms of either age or the rates of obesity, CAD, hypertension, atrial fibrillation, or COPD. This discrepancy in comparison with previous reports [1,32] may be due to a small sample size in our study, pre-CPET use of bronchodilators in some patients, and a relatively low proportion of patients with obesity (median BMI 27.5 kg/m^2^).

## 5. Conclusions

Among patients qualified for cardiac rehabilitation, patient’s sex, T2DM, CKD, and the CCI score emerged as key predictors of reduced exercise capacity. Reduced peak VO_2_ was more frequently observed in men, whereas an elevated VE/VCO_2_ slope was more prevalent in women, suggesting potential sex-specific physiological responses. These findings underscore the need for exercise assessment by CPET in all patients in order to design the best tailored rehabilitation approach, taking into account specific clinical profiles and functional limitations.

## Figures and Tables

**Figure 1 jcm-14-07512-f001:**
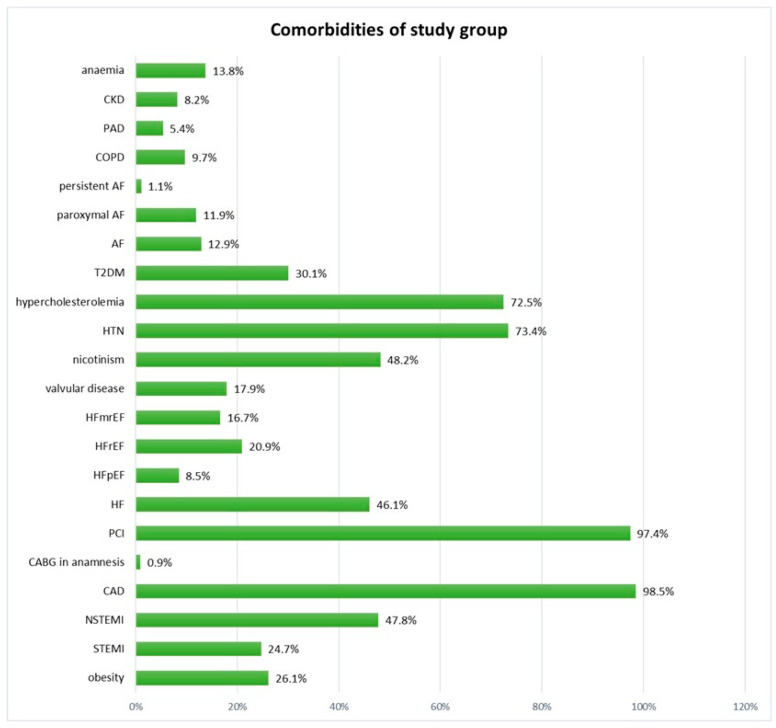
Comorbidities in the study group. Figure Legend: AF—atrial fibrillation; CAD—coronary artery disease; CABG—coronary artery bypass grafting; CKD—chronic kidney disease; COPD—chronic obstructive pulmonary disease; T2DM—type 2 diabetes mellitus; HF—heart failure; HFmrEF—heart failure with mid-range ejection fraction; HFpEF—heart failure with preserved ejection fraction; HFrEF—heart failure with reduced ejection fraction; HTN—hypertension; NSTEMI—non-ST-elevation myocardial infarction; PAD—peripheral artery disease; PCI—percutaneous coronary intervention; STEMI—ST-elevation myocardial infarction.

**Figure 2 jcm-14-07512-f002:**
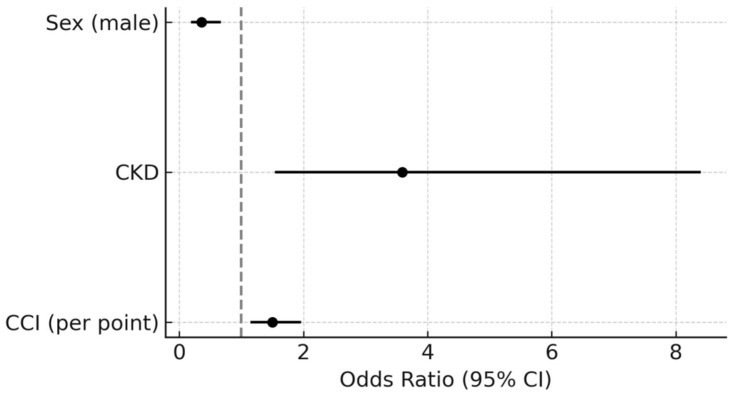
Independent predictors of VE/VCO_2_ slope ≥36.

**Figure 3 jcm-14-07512-f003:**
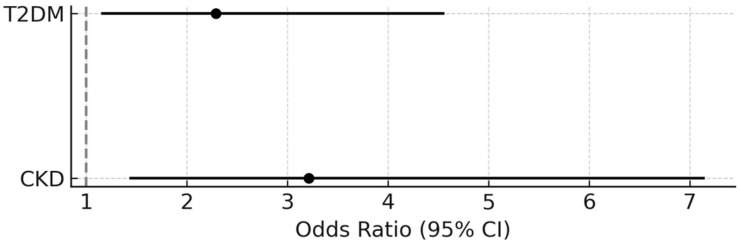
Independent predictors of peak VO_2_ < 70% of predicted value.

**Table 1 jcm-14-07512-t001:** Baseline characteristics of the study group.

	Study Group (n = 425) Median (Q1–Q3); n (%)
Age [years]	63 (56–70)
Sex [male]	316 (74.3)
Systolic blood pressure [mmHg]	130 (120–142)
Diastolic blood pressure [mmHg]	77 (70–84)
Heart rate [bpm]	69 (63–75)
BMI [kg/m^2^]	27.5 (25.3–30)
Charlson Comorbidity Index score [points]	4 (3–5)
**Symptoms and signs**	
Edema	23 (5.4)
Pulmonary circulatory congestion	33 (7.8)
Dyspnea on exertion	78 (18.4)
Dyspnea at rest	10 (2.4)
rapid fatigue during exercise	225 (52.9)
NYHA class 0	350 (82.4)
NYHA class I	3 (0.7)
NYHA class II	49 (11.5)
NYHA class III	23 (5.4)
Chest pain	45 (10.6)
CCS grade 0	401 (94.4)
CCS grade 1	2 (0.5)
CCS grade 2	19 (4.5)
CCS grade 3	3 (0.7)
Palpitations	32 (7.5)
**Echocardiography parameters**	
LVEF [%]	50 (42–60)
LVEF < 50%	175 (41.2)
LVDD grade 0	276 (65.4)
LVDD grade 1	117 (27.5)
LVDD grade 2	18 (4.2)
LVDD grade 3	4 (0.9)
LVDD grade 4	8 (1.9)
LVDD	147 (34.6)
**Treatment at admission**	
Statin	324 (76.2)
Statin with ezetimibe	93 (21.9)
ACE inhibitor/ARB/ARNI	393 (92.5)
Beta-blocker	374 (88.0)
CCB	128 (30.1)
Diuretic	194 (45.7)
MRA	123 (28.9)
Metformin	104 (24.5)
SGLT2 inhibitor	128 (30.1)
Amiodarone	28 (6.6)
**Exercise tests parameters**	
VE/VCO_2_ slope	31.2 (27.6–35.3)
Patients with VE/VCO2 slope ≥ 36	87 (20.8)
Peak VO_2_ [mL/min/kg]	17 (14–22)
Predicted peak VO_2_ [%]	78 (67–89)
Patients with peak VO_2_ < 70% of predicted	124 (29.4)
Patients with peak VO_2_ ≥ 85%	281 (66.6)
6MWD [m]	555 (488–611)
Predicted 6MWD [%]	99.8 (89.3–107.3)
Patients with 6MWD ≥ 100% of predicted	208 (50.4)
**Laboratory parameters**	
HGB [g/dL]	13.5 (12.6–14.4)
Creatinine [mg/dL]	1.0 (0.8–1.1)
eGFR [mL/min/1.73 m^2^]	87.1 (69.0–105.5)
eGFR > 90 mL/min/1.73 m^2^	191 (44.9)
eGFR ≥ 60 and <90 mL/min/1.73 m^2^	166 (39.1)
eGFR ≥ 45 and <60 mL/min/1.73 m^2^	41 (9.6)
eGFR ≥ 30 and <45 mL/min/1.73 m^2^	16 (3.8)
eGFR ≥ 15 and <30 mL/min/1.73 m^2^	3 (0.7)
Urea [mg/dL]	37 (30–44)
Total cholesterol [mg/dL]	120 (106–145)
Low-density lipoproteins [mg/dL]	63 (48–79)
High-density lipoproteins [mg/dL]	41 (36–50)
Triglycerides [mg/dL]	106 (83–139)
Glucose [mg/dL]	98 (91–110)

6MWD—six-minute walk distance; ACE—angiotensin-converting enzyme; ARB—angiotensin receptor blocker; ARNI—angiotensin receptor neprilysin inhibitor; BMI—body mass index; CCB—calcium channel blocker; CCS—Canadian Cardiovascular Society classification; eGFR—estimated glomerular filtration rate; HGB—hemoglobin; LVDD—left ventricular diastolic dysfunction; LVEF—left ventricular ejection fraction; MRA—mineralocorticoid receptor antagonist; NYHA—New York Heart Association classification; peak VO_2_—peak oxygen uptake during exercise; SGLT2—sodium–glucose cotransporter 2.

**Table 2 jcm-14-07512-t002:** Comparison between the subgroups: VE/VCO_2_ slope < 36 vs. VE/VCO_2_ slope ≥ 36 and between the subgroups: peak VO_2_ ≥ 70% of predicted vs. peak VO_2_ < 70% of predicted.

	VE/VCO_2_ Slope < 36 n = 331	VE/VCO_2_ Slope ≥ 36 n = 87	*p*-Value	Peak VO_2_ ≥ 70% of Predicted n = 298	Peak VO_2_ < 70% of Predicted n = 124	*p*-Value
Age [years]	62 (54–69)	69 (63–74)	0.210	64 (56–70)	63 (54–70)	0.0638
Sex [male]	256 (77.3)	55 (63.2)	0.007	211 (70.8)	102 (82.3)	0.014
Obesity [BMI > 30 kg/m^2^]	87 (26.6)	21 (24.1)	0.641	79 (26.7)	29 (23.8)	0.535
SBP [mmHg]	131 (120–143)	126 (114–137)	0.015	131 (120–144)	127 (113–138)	0.008
DBP [mmHg]	78 (70–84)	76 (69–84)	0.186	78 (70–84)	77 (68–84)	0.124
HR [bpm]	68 (62–72)	72 (67–80)	<0.0001	69 (63–74)	70 (63–78)	0.320
NYHA class 0	286 (86.4)	58 (66.7)	<0.0001	251 (84.2)	96 (77.4)	0.019
NYHA class I	3 (0.9)	0 (0.0)		3 (1.0)	0 (0.0)	
NYHA class II	30 (9.1)	18 (20.7)		34 (11.4)	15 (12.1)	
NYHA class III	12 (3.6)	11 (12.6)		10 (3.4)	13 (10.5)	
LVEF [%]	55 (45–60)	45 (36–52)	<0.0001	55 (45–60)	45 (38–56)	<0.0001
Patients with LVEF < 50%	113 (34.4)	57 (65.5)	<0.0001	103 (34.6)	72 (58.1)	<0.0001
HGB [g/dL]	13.7 (12.8–14.5)	12.9 (12.0–14.4)	0.002	13.6 (12.7–14.6)	13.3 (12.3–14.2)	0.042
eGFR [mL/min/1.73 m^2^]	92 (76–110)	71 (57–86)	0.0002	90 (71–106)	81 (64–103)	0.0003
Charlson Comorbidity Index score [points]	3 (2–4)	5 (4–6)	<0.0001	3 (2–3)	4 (3–6)	0.0004
Coronary artery disease	324 (97.9)	86 (98.9)	0.559	294 (98.7)	120 (96.8)	0.196
Heart failure	123 (37.2)	66 (75.9)	<0.0001	113 (37.9)	82 (66.1)	<0.0001
Hypertension	238 (77.5)	69 (79.3)	0.164	216 (72.5)	94 (75.8)	0.481
Type 2 diabetes mellitus	83 (25.1)	42 (48.3)	<0.0001	76 (25.5)	52 (41.9)	0.0008
Atrial fibrillation	42 (12.7)	13 (14.9)	0.580	35 (11.7)	20 (16.1)	0.223
Chronic obstructive pulmonary disease	30 (9.1)	11 (12.6)	0.318	26 (8.7)	15 (12.1)	0.287
Peripheral artery disease	12 (3.6)	10 (11.5)	0.003	10 (3.4)	13 (10.5)	0.003
Chronic kidney disease	14 (4.2)	20 (23.0)	<0.0001	12 (4.0)	23 (18.6)	<0.0001
Valvular disease	43 (13.0)	31 (35.6)	<0.0001	39 (13.1)	37 (29.9)	<0.0001
LVDD	97 (20.3)	45 (51.7)	<0.0001	90 (30.2)	57 (46.0)	0.002
Anemia	35 (10.8)	19 (23.2)	0.003	33 (11.3)	24 (20.2)	0.019

Data presented as median (IQR) and n (%). BMI—body mass index; DBP—diastolic blood pressure; eGFR—estimated glomerular filtration rate; HGB—hemoglobin; HR—heart rate; LVDD—left ventricular diastolic dysfunction; LVEF—left ventricular ejection fraction; NYHA—New York Heart Association classification; SBP—systolic blood pressure.

**Table 3 jcm-14-07512-t003:** Predictors of VE/VCO_2_ slope ≥36 and peak VO_2_ <70% of predicted value (logistic regression).

	VE/VCO_2_ Slope ≥36						Peak VO_2_ <70% of Predicted Value					
	Univariate			Multivariate			Univariate			Multivariate		
	OR	95% CI	*p*-Value	OR	95% CI	*p*-Value	OR	95% CI	*p*-Value	OR	95% CI	*p*-Value
Sex (male)	0.50	0.30–0.84	0.079	0.36	0.19–0.67	0.001	1.91	1.13–3.22	0.015	-	-	-
LVEF < 50%	3.67	2.23–6.02	<0.0001	-	-	-	2.62	1.71–4.03	<0.0001	-	-	-
HF	5.32	3.10–9.11	<0.0001	-	-	-	3.2	2.06–4.96	<0.0001	-	-	-
T2DM	2.79	1.71–4.55	<0.0001	-	-	-	2.11	1.36–3.28	0.0009	2.29	1.15–4.56	0.018
PAD	3.45	1.44–8.29	0.006	-	-	-	3.73	1.44–7.92	0.005	-	-	-
CKD	6.76	3.25–14.10	<0.0001	3.59	1.54–8.40	0.003	5.43	2.61–11.31	<0.0001	3.21	1.43–7.15	0.004
Valvular disease	3.71	2.15–6.38	<0.0001	-	-	-	2.82	1.69–4.71	0.0001	-	-	-
LVDD	2.59	1.60–4.19	0.0001	-	-	-	1.97	1.28–3.03	0.002	-	-	-
Anemia	2.49	1.34–4.63	0.004	-	-	-	1.98	1.11–3.51	0.021	-	-	-
CCI score (per point)	1.67	1.46–1.97	<0.0001	1.5	1.15–1.96	0.003	1.26	1.12–1.41	0.0001	-	-	-

CCI–Charlson Comorbidity Index; CKD–chronic kidney disease; T2DM–type 2 diabetes mellitus; HF–heart failure; LVDD–left ventricular diastolic dysfunction; LVEF–left ventricular ejection fraction; PAD–peripheral artery disease.

## Data Availability

The datasets presented in this article are not readily available because the data is covered by traditional medical confidentiality. Requests to access the datasets should be directed to the Military Institute of Medicine—National Research Institute.

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
