# Peer review of "Predictors of Impaired Exercise Performance in Patients Qualified for Cardiac Rehabilitation: The Impact of Sex and Comorbidities"

_jcm, 2025, doi:10.3390/jcm14217512_

Round 1

Reviewer 1 Report

Comments and Suggestions for Authors

Whilst this is an interesting study and would have taken a significant amount of time to review and analyse. I believe the authors need to review the document and resubmit a more succinct article, ensuring that all aspects are taken care of. Currently it is a little disjointed and needs various information added. 

Author Response

REVIEWER 1 - COMMENTS

Reviewer’s comment:

“What is not inferenced is the problem with cardiac rehabilitation? What is the problem with the current processes in place?”

Answer:

Thank you for this comment. Comprehensive CR leads to a significant reduction in hospitalizations, adverse nonfatal cardiovascular events, and mortality rates, as well as improved cardiovascular risk profiles and exercise capacity in patients with cardiovascular disease [Hansen 2020]. Despite numerous studies, there are no clear recommendations for design of the exercise programme, e.g. on percentage of the VO2peak, HRpeak, HRR, Wpeak or HR zone between VT1 and VT2 [hansen 2020]. Patients with heart disease have numerous comorbidities that influence exercise capacity and prognosis. The challenge remains to create training during CR in patients with comorbidities, e.g. diabetes mellitus [kosinski 2019], chronic kidney disease (CKD) [Hayden 2024] or atrial fibrillation (Cersosimo 2025). Having this in mind, the main problem with current cardiac rehabilitation processes is the limited individualization of training programs. Existing protocols often do not sufficiently consider the impact of comorbidities and sex-related physiological differences on exercise capacity. Our study addresses this issue by identifying key predictors such as type 2 diabetes, chronic kidney disease, and sex, which may support a more tailored approach to rehabilitation and improve clinical outcomes. To clarify the issue, we have added an explanation in the introduction section - page 2, 1. section: Introduction.

Hansen D, Abreu A, Ambrosetti M, Cornelissen V, Gevaert A, Kemps H, Laukkanen JA, Pedretti R, Simonenko M, Wilhelm M, Davos CH, Doehner W, Iliou MC, Kränkel N, Völler H, Piepoli M. Exercise intensity assessment and prescription in cardiovascular rehabilitation and beyond: why and how: a position statement from the Secondary Prevention and Rehabilitation Section of the European Association of Preventive Cardiology. Eur J Prev Cardiol. 2022 Feb 19;29(1):230-245. doi: 10.1093/eurjpc/zwab007. Erratum in: Eur J Prev Cardiol. 2024 Sep 20;31(13):e102. doi: 10.1093/eurjpc/zwad397. PMID: 34077542.

Kosinski C, Besson C, Amati F. Exercise Testing in Individuals With Diabetes, Practical Considerations for Exercise Physiologists. Front Physiol. 2019 Sep 27;10:1257. doi: 10.3389/fphys.2019.01257. PMID: 31611821; PMCID: PMC6777138.

Cersosimo A, Longo Elia R, Condello F, Colombo F, Pierucci N, Arabia G, Matteucci A, Metra M, Adamo M, Vizzardi E, LA Fazia VM. Cardiac rehabilitation in patients with atrial fibrillation. Minerva Cardiol Angiol. 2025 Aug 1. doi: 10.23736/S2724-5683.25.06885-1. Epub ahead of print. PMID: 40748298

Reviewer’s comment:

"1) In the study population section, the numbers of participants do not add up.

The study identified 518 patients. Excluded 51 after cardiothoracic surgery, and 42 who did not undergo CPET – leaving 425 for the study.

The subgroup of good exercise capacity (n= 298), and reduced exercise capacity (n = 124), this leaves 3 patients (n = 422) in an unidentified group. Is it possible to identify where these patients fit.

Equally the VE/VCO2 slope >36 (n = 87) and <36 (n = 331) = total (n=418). Please could the authors clarify what group the other 6 patients belong too."

Table 2: there are missing data as there should be 425 patients

Answer: A retrospective analysis was performed based on medical records. Differences resulted from missing data for predicted peak VO2 [%] (no body mass/BMI) and missing data for VE/VCO2 slope >36, but these did not involve the same patients, hence the discrepancy with the total sample size. Due to the lack of data, it was impossible to assign 3 patients to the comparative analysis according to peak VO2 and 7 patients to the analysis according to VE/VCO2 slope. This did not affect the statistical results, but we included the information in the limitation section 4.3.- page 14

Reviewer’s comment:

In section 2.3 Cardiopulmonary exercise test

Peak VO2 2 needs to be subscript..

Answer: It was corrected - page 4

Reviewer’s comment:

The authors mention that they used Hansen and Jones formula for predicting peak VO2. These are two different formulae – The reference does not correlate with either formula. Here are the actual references.

  • Jones NL, Makrides L, Hitchcock C, Chypchar T, McCartney N. Normal standards for an

incremental progressive cycle ergometer test. American Review of Respiratory Disease.

1985;131(5):700–708. doi: 10.1164/arrd.1985.131.5.700.

  • Hansen JE, Sue DY, Wasserman K. Predicted values for clinical exercise testing. American

Review of Respiratory Disease. 1984;129(2):S49–S55. doi: 10.1164/arrd.1984.129.2P2.S49."

Answer: We thank the Reviewer for this important methodological remark. It was corrected - page 5, in section 2.3 Cardiopulmonary exercise test.

Reviewer’s comment:

The authors should reference the methodology used for calculating the VE/VCO2 slope.

Answer:

Reference was added - page 5, section 2.3. and page 16, section References.

We used the CPE results from the patient's records. We had no influence on the method used to determine the VE/VCO2 slope. The VE/VCO2 slope calculation method set by the ergospirometer manufacturer (the CORTEX METALYZER® 3B and analyzed in MetaSoft Studio) was used. VE/VCO₂ slope was computed by ordinary least squares linear regression of VE (y) on VCO₂ (x) over the exercise interval from ~1 min after onset to the end of the isocapnic buffering phase (≈ respiratory compensation point).

Reviewer’s comment:

The authors have a section 2.4. Six-minute walk test.

It is possible to the authors to elaborate on why the 6-minute walk test was also included?

Answer: The 6MWT is a simple and inexpensive test that is well-tolerated by the patient and widely used. Most rehabilitation centers assess exercise capacity not based on the CPET, but rather on a classic exercise test and the 6MWT. We've included data from the 6MWT along with the CPET so that physicians, who don't have the CPET, can apply the data to their daily practice.

The median (Q1-Q3) predictive value of the achieved 6MWT distance was 99.8 (89.3–107.3)%, significantly higher than the median predictive value of VO2 78 (67–89)%. On the other hand, fewer patients had a 6MWT ≥100% (208), than had a VO2 >85% (281). We assumed that presenting data on 6MWT would better characterize our study group. We would appreciate accepting this approach.

It was added to section 3.1. Baseline characteristics, page 6.

Reviewer’s comment:

If is a reference value of some kind? It seems out of place.

Answer: There are several formulas for calculating the predictive value in healthy individuals - according to Table 1of the Giannitsi et al [Giannitsi 2019]

The 6-minute walking test (6MWT) is used to the assessment of daily activities performance as submaximal exercise tests. Therefore, we accepted the value of 100% predictive value according to the Enright formula as the correct value. Next we calculate percentage of preducted value.

6MWT distance correlate with symptoms e.g. "NYHA class II–IV and 6MWD (mean values ~400 m, 320 m and 225 m, respectively for NYHA class II, III and IV" [Yap 2015] and with prognosis, e.g. ⩽ 300 m is indicative of poor prognosis in HF [Alahdab 2009, Guazzi 2009], in stable coronary artery desease [Beatty 2012] and "cutoff distance for MACE was 392 m, with sensitivity of 76% and specificity of 53%" [Sohn 2025]. We do not use absolute values due to the good performance of the patients in the entire group and the lack of very advanced diseases.

Giannitsi S, Bougiakli M, Bechlioulis A, Kotsia A, Michalis LK, Naka KK. 6-minute walking test: a useful tool in the management of heart failure patients. Ther Adv Cardiovasc Dis. 2019 Jan-Dec;13:1753944719870084. doi: 10.1177/1753944719870084. PMID: 31441375; PMCID: PMC6710700.

Yap J, Lim FY, Gao F, et al. Correlation of the New York heart association classification and the 6-minute walk distance: a systematic review. Clin Cardiol 2015; 38: 621–628.

Alahdab MT, Mansour IN, Napan S, et al. Six minute walk test predicts long-term all-cause mortality and heart failure rehospitalization in African-American patients hospitalized with acute decompensated heart failure. J Card Fail 2009; 15: 130–135.

Guazzi M, Dickstein K, Vicenzi M, et al. Six-minute walk test and cardiopulmonary exercise testing in patients with chronic heart failure: a comparative analysis on clinical and prognostic insights. Circ Heart Fail 2009; 2: 549–555.

Beatty AL, Schiller NB, Whooley MA. Six-minute walk test as a prognostic tool in stable coronary heart disease: data from the heart and soul study. Arch Intern Med. 2012 Jul 23;172(14):1096-102. doi: 10.1001/archinternmed.2012.2198. PMID: 22710902; PMCID: PMC3420342.

Sohn, S., Jeon, J., Lee, J.E. et al. Prognostic value of the six-minute walk test in patients with cardiovascular disease. Sci Rep 15, 20817 (2025). https://doi.org/10.1038/s41598-025-04480-9

Reviewer’s comment:

Enright is reference number 19 not 20, please revise.

Answer: It was corrected - page 5, section Six-minute walk test

Reviewer’s comment:

What was the timeframe from CPET to six minute walk test? It is not clear how long the rest period was in between the two tests.

Answer: We used a second, repeated 6MWT. There were 2-3 days between the 6MWT and CPET. The 6MWT and CPET were not performed on the same day. It was added in section Six-minute walk test, page 5

Reviewer’s comment:

Statistical analysis

In this section, the authors have used several different statistics that correspond to the data collected. The authors have clearly defined how they will be splitting up the data based on exercise capacity and ventilatory capacity.

Answer: Thank you for this positive comment.

Reviewer’s comment:

This still has the manuscript details on what should be included at the start… Please remove this and start with a paragraph that replaces this.

“This section may be divided by subheadings. It should provide a concise and precise description of the experimental results, their interpretation, as well as the experimental conclusions that can be drawn

Answer: It was corrected.

Reviewer’s comment:

The authors state that 281 patients had a normal exercise tolerance. What does this mean and what is it based off?

  1. Answer: Performance was assessed using CPET, based on the achieved peak VO2 and its predictive value. Normal exercise tolerance was defined according to Glaab et al 2022. Table 1 was corrected, page 8, and page 6.
  2. Glaab, T.; Taube, C. Practical guide to cardiopulmonary exercise testing in adults. Respir Res. 2022, 23, 9. doi: 10.1186/s12931-021-01895-6.

Reviewer’s comment:

The authors report that 50.4 patients achieved a 6MWT above predicted values – 555m (median distance 488- 611m) Did medication impact the exercise tolerance results?

Answer:

6MWT provides information only about submaximal effort. On the one hand, several studies have shown moderate-to-strong correlations of 6MWD with peak aerobic capacity (peak VO2) in CPET 2 [Giannitsi et al]. We do not use 6MWT to determine capacity. Moreover, the discrepancy in the number of patients who achieve >100% and the median predictive value of 6MWT and peakVO2 emphasizes that these tests cannot be treated interchangeably.

We assessed exercise capacity by CPET in patients with cardiovascular disease fully treated with recommended medications, including beta-blockers, ACE inhibitors/ARBs/ARNI, diuretics, and statins. In particular, beta-blockers may reduce maximal heart rate and, consequently, peak VO₂, while having less impact on submaximal tests such as the 6MWT. However, since almost all patients received similar evidence-based therapy, the effect of medications on between-group differences in exercise performance was likely minimal.

Giannitsi S, Bougiakli M, Bechlioulis A, Kotsia A, Michalis LK, Naka KK. 6-minute walking test: a useful tool in the management of heart failure patients. Ther Adv Cardiovasc Dis. 2019 Jan-Dec;13:1753944719870084. doi: 10.1177/1753944719870084. PMID: 31441375; PMCID: PMC6710700.

Palau, P, Seller, J, Domínguez, E. et al. Effect of β-Blocker Withdrawal on Functional Capacity in Heart Failure and Preserved Ejection Fraction. JACC. 2021 Nov, 78 (21) 2042–2056.https://doi.org/10.1016/j.jacc.2021.08.073

Effects of metoprolol CR in patients with ischemic and dilated cardiomyopathy : the randomized evaluation of strategies for left ventricular dysfunction pilot study. Circulation. 2000 Feb 1;101(4):378-84. doi: 10.1161/01.cir.101.4.378. PMID: 10653828.

Reviewer’s comment:

Below table 2: The grammar needs to be fixed… “In In coIn comparison with patients with a peak peak VO2 ≥ 70%, the subgroup with VO2 of < 70% comprised a greater”

Answer: It was corrected - page 10.

Reviewer’s comment:

Of note was that males were more likely to exhibit reduced VO2 peak, whilst women more frequently presented with ventilatory inefficiency. – What do the authors think contributes to this difference?

Answer:

Thank you for this suggestion. The VE/VCO2 slope reflects the slope in the ratio between VE and VCO2. In female, a higher VE/VCO2 slope was more often observed than reduced peakVO2. We hypothesize that in female, ventilation increases more during exercise than there is difficulty exhaling CO2, which may be related to factors such as emotions or anxiety. In men, the VE/VCO2 slope reflects the cause of the reduced peak VO2, a problem exhaling CO2, for example, due to congestion in the pulmonary circulation. Future research in this area is necessary.

It was added in section 4.1. Exercise capacity parameters and sex, page 13

Reviewer’s comment:

4.1. Exercise capacity parameters and sex

VO2 should read VO2 There are other sections throughout the discussion where this change should be made.

Answer: They were corrected, page 12.

Reviewer’s comment:

There was no discussion of six-minute walk test in the discussion and I am still left not understanding as to how these fits with this study.

Answer: The discussion section was supplemented with information also discussed in previous responses to the reviewer, page 12, section 4 Discussion.

Reviewer’s comment:

In conclusion the authors suggest that the findings underscore the need to holistic and individualised approach to exercise training in cardiac rehab tailored to the patients’ specific needs – I think it would be pertinent to mention here that the authors recommend that all patients should have cardiopulmonary exercise assessment to ascertain the best tailored rehab approach….

Answer: Thank you for appreciating our conclusions, we will strengthen the message in the 5. Conclusions, page 14.

Reviewer’s comment:

Although this is suggested, the authors still do not document anything about cardiac rehab and the different approaches for each of their groups which present a dichotomy.

Reviewer’s comment:

Whilst this is an interesting study and would have taken a significant amount of time to review and analyse. I believe the authors need to review the document and resubmit a more succinct article, ensuring that all aspects are taken care of.

Answer: Thank you very much for all your constructive comments. We have carefully addressed each point and incorporated the suggested changes. We believe that these revisions have improved the overall clarity and quality of the manuscript.

Reviewer 2 Report

Comments and Suggestions for Authors

Congratulations to the authors for their relevant manuscript; I just have some comments about it:

The decision to exclude post-cardiac surgery patients appears appropriate; however, the justification provided is insufficient and should be clarified.

Chronic kidney disease was defined as eGFR <60 or a prior diagnosis, although no information on the distribution of CKD stages is presented.

For the multivariate regression variables were selected based on a univariate threshold of p <0.10, which may predispose the model to overfitting. It would be important to indicate whether variance inflation factors were assessed to address potential collinearity.

Logistic regression models assume a linear relationship between continuous predictors and the log-odds of the outcome.

The discussion would benefit from elaboration on how the identified predictors can be practically incorporated into individualized cardiac rehabilitation strategies. Moreover authors are encouraged to include in their discussion other examples of cardiac rehabilitation (doi: 10.23736/S2724-5683.25.06885-1) in order to increase its validity

Authors should consider adding additional visualizations, such as forest plots of odds ratios with confidence intervals.

Author Response

Reviewer’s comment:

Congratulations to the authors for their relevant manuscript; I just have some comments about it:

Answer: Thank you for this positive comment.

Reviewer’s comment:

The decision to exclude post-cardiac surgery patients appears appropriate; however, the justification provided is insufficient and should be clarified.

Answer: We excluded 51 patients after cardiothoracic surgery, due to an acute, temporary condition after surgery, e.g. pleural effusion or restricted breathing due to pain in the sternum. Our aim was to assess exercise capacity without transitional states. The postoperative period of up to several months can significantly impact performance and interfere with analysis results of impact chronic diseases.

  1. Materials and Methods - Study population was completed Page 4

Reviewer’s comment:

Chronic kidney disease was defined as eGFR <60 or a prior diagnosis, although no information on the distribution of CKD stages is presented.

Answer: Thank you for this suggestion. The data was completed in Table 1, page 8

Reviewer’s comment:

For the multivariate regression variables were selected based on a univariate threshold of p <0.10, which may predispose the model to overfitting.

Answer: We appreciate the Reviewer’s comment regarding the variable selection strategy for the multivariate regression model. We acknowledge that selecting variables based on a univariate threshold of p < 0.10 may, in some cases, increase the risk of model overfitting. However, this approach is widely accepted in clinical and epidemiological studies aiming to identify independent predictors rather than to construct predictive models. A liberal threshold (p < 0.10–0.15) allows the inclusion of potentially relevant variables that may not reach strict statistical significance due to sample size (examplary ref Source Code Biol Med. 2008 Dec 16;3:17). All variables entered into the multivariate model were clinically meaningful and supported by prior literature. Therefore, we believe that our selection strategy provided a balanced compromise between statistical rigor and clinical interpretability.

Reviewer’s comment:

It would be important to indicate whether variance inflation factors were assessed to address potential collinearity.

Answer: Collinearity diagnostics (VIF, variance inflation factors) were performed for all candidate variables before entering them into multivariate models. The VIF values were < 2.0, indicating no relevant collinearity, besides HF and left ventricular ejection fraction (LVEF < 50%) which were borderline moderately correlated (VIF = 2.10). However, both were retained in the multivariate model due to their distinct clinical meaning — HF reflecting a syndromic diagnosis and LVEF serving as an objective echocardiographic parameter with well-established prognostic value. We would appreciate accepting this  clinically justified approach.

It was added in section Statistical analysis, page 5.

Reviewer’s comment:

Logistic regression models assume a linear relationship between continuous predictors and the log-odds of the outcome.

Answer: We thank the Reviewer for this important methodological remark. To minimize the risk of violating this assumption, all key continuous parameters were categorized based on clinically meaningful thresholds (e.g., LVEF <50%, anemia - hemoglobin <12 g/dL).

Reviewer’s comment:

The discussion would benefit from elaboration on how the identified predictors can be practically incorporated into individualized cardiac rehabilitation strategies. Moreover authors are encouraged to include in their discussion other examples of cardiac rehabilitation (doi: 10.23736/S2724-5683.25.06885-1) in order to increase its validity

Answer: Thank you for this suggestion. Discussion was completed, page 14.

Reviewer’s comment:

Authors should consider adding additional visualizations, such as forest plots of odds ratios with confidence intervals.

Answer: Thank you for this suggestion. The plots are presented on Figures 2 and 3, page 10 and 11 for Figure 2 and page 11 i 12 for Figure 3

Round 2

Reviewer 2 Report

Comments and Suggestions for Authors

Congratulations to the authors for the revised version of their manuscript.